# Digital assessment of walking ability: Validity and reliability of the automated figure-of-eight walk test in older adults

Hyun-Ho Kong[1,2*☉], Kwangsoo Shin[3], Dong-Seok Yang[4], Hye-Young Gu[1], Hyeon-Seong Joo[5], Hyun-Chul Shon[6,7*☉]

1 Department of Rehabilitation Medicine, Chungbuk National University Hospital, Cheongju, Republic of Korea, 2 Department of Rehabilitation Medicine, Chungbuk National University College of Medicine, Cheongju, Republic of Korea, 3 Graduate School of Public Health and Healthcare Management, Songeui Medical Campus, The Catholic University of Korea, Seoul, Republic of Korea, 4 Technology Strategy Center, Neofect, Seongnam, Republic of Korea, 5 Department of Physical Therapy, Daejeon University, Daejeon, Republic of Korea, 6 Department of Orthopaedic Surgery, Chungbuk National University Hospital, Cheongju, Republic of Korea, 7 Department of Orthopaedic Surgery, Chungbuk National University College of Medicine, Cheongju, Republic of Korea

☉ These authors contributed equally to this work.
* jimlight@hanmail.net (HHK); hyunchuls@chungbuk.ac.kr (HCS)

## Abstract

### Background

The Figure-of-Eight Walk Test (F8WT) can assess straight- and curved-path walking ability, but the validity and reliability of automated measurement of the F8WT using digital device has not yet been studied. The aim of this study was to verify the validity (method comparison) and test-retest reliability of the automated FW8T (aFW8T) using a digital device based on image analysis by comparing the results of the aF8WT with those of the manual F8WT (mF8WT).

### Methods

Community-dwelling older adults underwent the mF8WT performed by a physiotherapist and the aF8WT using the Digital Senior Fitness Test system. To verify the test-retest reliability, the aF8WT was administered again to a randomly selected group of participants one week after the baseline test. The intraclass correlation coefficient (ICC) and Pearson's correlation analysis were used to verify the degree of agreement between the results of and correlation between the mF8WT and aF8WT, respectively. The 95% confidence interval (CI) of the limits of agreement (LoA) was obtained using Bland–Altman analysis.

### Results

The analysis included 83 participants (mean age 71.6 ± 4.7 years). The participants' mF8WT and aF8WT results were 29.1 ± 4.9 and 29.8 ± 4.9 seconds, respectively. Pearson's correlation analysis showed a very strong correlation between the mF8WT and aF8WT results with r = 0.91 (p < 0.001), and the ICC between the mF8WT and aF8WT results was 0.95 (0.91–0.97), showing excellent agreement. The 95% CI of the LoA was

**Data availability statement:** Our data cannot be shared publicly due to ethical and legal restrictions, as the data contain potentially identifying or sensitive patient information. The Institutional Review Board at Chungbuk National University Hospital has prohibited the sharing of even de-identified data to ensure patient confidentiality and data protection. Researchers who meet the criteria for accessing confidential data may contact the Chungbuk National University Hospital Ethics Committee for data requests. The contact information is as follows: Data cannot be shared publicly because of ethical and legal restrictions, as the data contain potentially identifying or sensitive patient information. Data are available from the Chungbuk National University Hospital Ethics Committee (contact via cbnuhirb@naver.com, TEL: +82-43-269-6771, Chungbuk National University Hospital Ethics Committee, 776 1Sunhwan-ro, Seowon-gu, Cheongju, Chungbuk, Republic of Korea) for researchers who meet the criteria for access to confidential data.

**Funding:** This research was supported by the Medical Device Technology Development Program (grant number: 20014701, modular quantitative aging assessment and care service based on multiple sensors for aging in-home) funded by the Ministry of Trade, Industry, and Energy (MOTIE, Sejong, Republic of Korea). The funders had no role in study design, data collection and analysis, decision to publish, or preparation of the manuscript.

**Competing interests:** The authors have declared that no competing interests exist.

-0.7 (-4.8 to 3.3) seconds in the Bland–Altman analysis. In an analysis of the test-retest reliability of the aF8WT, participants' aF8WT results were 30.9±4.7 seconds (baseline) and 29.6±4.9 seconds (retest), with an ICC of 0.94 (0.81–0.98, $p < 0.001$), indicating excellent reliability.

## Conclusion

Automated measurement of the F8WT using a digital device showed excellent validity and reliability. The aF8WT can be used to assess and monitor the walking ability of community-dwelling older adults.

## Introduction

Walking is a complex motor task that requires interactions between the body systems involved in movement (the musculoskeletal and cardiopulmonary systems) and the nervous system (brain and peripheral nerves) [1,2]. Older adults show changes in gait parameters, such as lower walking speeds, shorter step lengths, and a higher percentage of double-limb support than younger adults [3,4]. A decline in walking ability is associated with an increased risk of falls and fractures in older adults, hospitalization, and mortality, and decreased ability to perform activities of daily living (ADLs) and quality of life [5–8].

Physical function tests to measure walking ability in older adults include the short-distance (4, 6, and 10 m) walk test and Timed Up and Go (TUG) test, which are commonly performed in clinical settings [9,10]. However, these tests cannot assess walking ability on curved-paths, which older adults must perform in their actual ADLs, because these tests only consist of a straight-path or single turn [11,12]. Walking on a curved path requires more complex motor skills than walking on a straight-path because it requires the ability to maintain mediolateral balance while controlling an asymmetrical walking pattern between the inner and outer legs [13]. In addition, curved-path walking is associated with cognitive functions, such as executive function [14].

The figure-of-eight walk test (F8WT) is a physical function test that measures the actual walking ability required in daily living, including curved-path walking (e.g., walking around obstacles, turning corners, and dancing) [13,15]. Although this test has been performed in slightly different ways in previous studies, it is commonly administered by walking along a straight- and curved-path consisting of two circles in the clockwise and counterclockwise directions [15–17]. F8WT results are significantly correlated with gait parameters, such as gait speed, gait variability of step length, and step width, and have high test-retest reliability [15,17]. In addition, F8WT results are highly correlated with walking ability in patients with stroke or Parkinson's disease [18,19], older adults after knee replacement surgery [20], and balance performance in older adults [16], and can be used to predict the risk of falls in older adults [21].

Recent advances in digital healthcare technology have led to attempts to measure traditional clinical physical functions using digital devices. For example, the Short Physical Performance Battery (SPPB) using a light detection and ranging sensor [22] or the TUG test using a smartphone or depth camera have been used in previous studies to measure walking ability in community-dwelling older adults or patients with certain medical conditions [23,24]. However, the SPPB and TUG tests are limited as they cannot evaluate walking along curved-paths. Despite our thorough literature search, no study has evaluated the validity and reliability of automated measurement of F8WT using a digital device yet. Using a digital device to measure the F8WT can minimize inter-rater reliability issues commonly encountered in manual

testing, as it ensures high reliability even when administered by individuals who are not trained technicians. In addition, a digital device enables automatic, real-time data storage and analysis, enhancing data traceability and integrity while reducing the need for manual data entry and error checking by researchers.

We developed a Digital Senior Fitness Test (DSFT) system that measures multiple domains of physical function in older adults using a digital device based on multiple sensors. This study aimed to verify the validity (method comparison) and test-retest reliability of the F8WT using the DSFT system by comparing the results of automated F8WT (aF8WT) performed using the DSFT system with those of manual F8WT (mF8WT).

## Methods

### Study population

This study targeted adults aged 65 years or older living in the community who were prospectively recruited from November 20, 2022 to July 15, 2023. The Korean version of the Fatigue, Resistance, Ambulation, Illnesses, and Loss of weight (K-FRAIL) scale was administered to assess the physical frailty status of the participants [25]. Information on the participants, such as age, sex, and body mass index (BMI), was collected.

The inclusion criteria were as follows: i) men and women aged 65 years or older living in the community, ii) those who understood the purpose of the study and voluntarily consented, and iii) those with all the data needed for analysis. Individuals with underlying medical conditions (such as cardiopulmonary/cerebrovascular disease, amputations, and severe arthritis) or sensory deficits (visual or hearing impairment) that made it difficult to perform the aF8WT or those who could not complete the test owing to personal reasons were excluded from the analysis.

A total of 92 participants were recruited, of whom 83 were finally included in the analysis, after excluding those who were unable to complete the test owing to underlying medical conditions (n = 5), hearing impairment that made it difficult to hear the auditory instructions provide by the DSFT system (n = 2), or unfamiliarity with testing using a digital device, leading them to abandon the test for personal reasons (n = 2). All 83 participants included in this study were able to safely perform the aF8WT without a supervisor. Ethical approval was obtained from the Institutional Review Board (IRB) of Chungbuk National University Hospital (IRB number: 2022-08-004-011), and written informed consent was obtained from all participants.

### Manual measurement of the F8WT

The mF8WT in this study was administered by a professional physiotherapist according to the protocol defined in the "National Fitness 100," a Senior Fitness Test (SFT) administered in Korea [26]. This protocol includes sitting on and standing up from a chair. First, a rectangle of 3.6 meters long and 2.4 meters wide was marked on the floor, and cones were placed on the back two corners of the rectangle (along the length). A chair was placed for the test participant at a distance of 2.4 meters from the cones on each side, and the test participant sat on the chair facing forward. At the start signal, the participant rose from the chair, walked as fast as possible in a clockwise direction around the cone placed at the back right, returned to the chair, and sat down. Then, they immediately stood up and walked counterclockwise around the left rear cone, returned to the chair, and sat down, repeating this process twice in succession (Fig 1). The test was administered after one or two practice sessions, and the test was administered twice with an interval of at least 10 minutes, the earlier of which was defined as the result of the mF8WT.

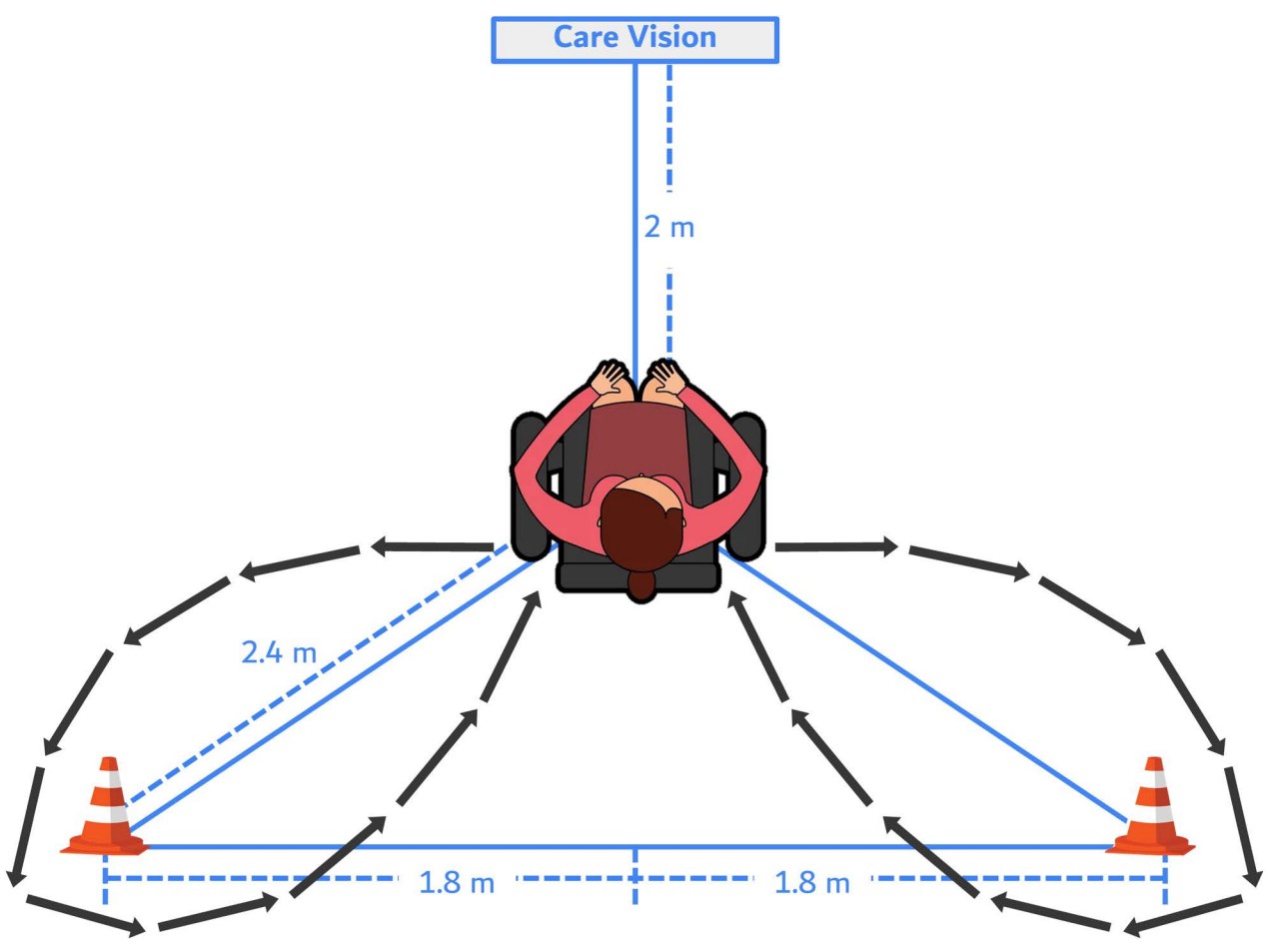

**Fig 1. Schematic of the method of the manual and automated measurement of the figure-of-eight walk test.**

### Development of the DSFT system

The DSFT system evaluates the SFT, which comprehensively assesses multiple domains of physical function (strength, flexibility, balance, and cardiopulmonary endurance) in older adults as a single digital device. The DSFT system consists of the following components: i) Care-Vision: kinematic data related to the participant's movements captured by a depth camera (Azure Kinect; Microsoft Corp., Redmond, WA, USA) are automatically analyzed by our custom artificial intelligence algorithm, which detects and evaluates key movement metrics and provides immediate results following each measurement; ii) Care-Pad: measures balance function by detecting the participant's foot pressure using a force plate installed with load cells; and iii) Care-Grip, an electronic hand dynamometer that can measure hand grip strength (Fig 2). The DSFT system is portable, requires no set-up time, and follows the same protocols as manual measurements, ensuring that the test duration is similar to that of manual assessments. Test results are analyzed and provided to the participant immediately upon completion of the test. The aF8WT performed in this study was measured using the Care-Vision.

### Automated measurement of F8WT

The aF8WT with the DSFT system provides pre-recorded visual and auditory instructions that can be repeated as many times as the participant desires before starting the test, thereby

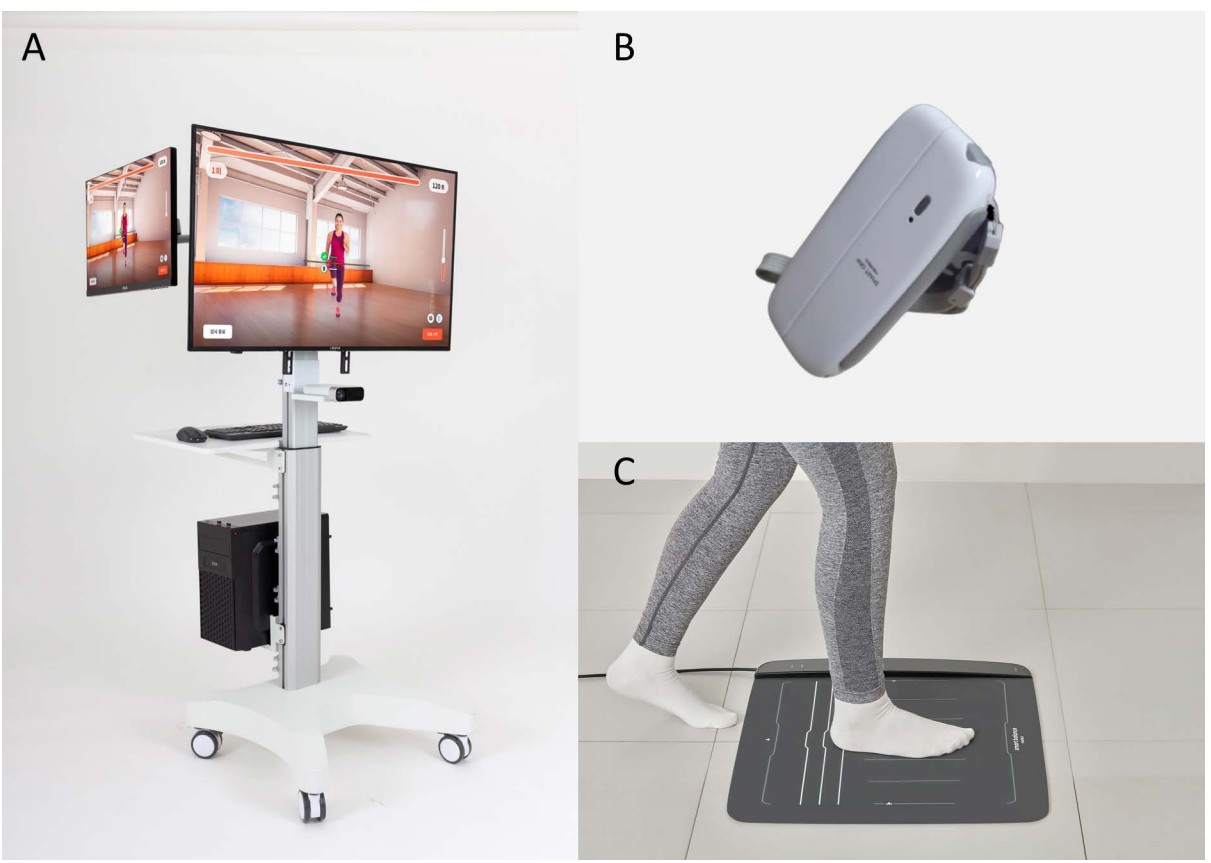

**Fig 2. Digital Senior Fitness Test System.** (A) Care-Vision(B) Care-Grip, and (C) Care-Pad.

allowing the participant to perform the test independently without the assistance of an assessor. The participants were also provided with the necessary visual and auditory instructions during the test. The method of performing the aF8WT was measured according to the manual measurement protocol described above. The chair in which the participant sat was placed 2 m in front of the DSFT system, and the cone used as the return point was placed at a distance of 2.4 m on either side of the chair (Fig 1). Prior to the start of the aF8WT, the DSFT system provided feedback by analyzing the skeletal position of the participant using a depth camera to ensure the correct starting position. To ensure that Care-Vision can accurately detect the participant's sitting and standing postures (detected by head height differences in the skeleton tracking of the depth images), the participant sat and stood in front of Care-Vision once before performing the aF8WT. In addition, to measure test-retest reliability, the aF8WT was repeated one week after the baseline test in 27 randomly selected participants.

## Statistical analysis

Continuous variables are presented as means and standard deviations, and categorical variables are presented as numbers and proportions. The intraclass correlation coefficient (ICC) $_{(2,1)}$, a two-way random, single measure with absolute agreement, was evaluated to verify the degree of agreement between the results of the mF8WT by the assessor and the aF8WT using the DSFT system, ensuring consistency between the two measurement methods [27]. Additionally, an ICC analysis was performed to assess the test-retest reliability of

the aF8WT, evaluating its consistency across repeated tests [28]. An ICC value of less than 0.40 was considered poor agreement, between 0.40 and 0.59 as fair, between 0.60 and 0.74 as good, and between 0.75 and 1.00 as excellent [29]. In addition, Bland–Altman analysis was performed to assess the bias and limits of agreement (LoA) between the mF8WT and aF8WT results, including the calculation of the 95% confidence interval (CI) for the LoA, to provide a detailed evaluation of measurement differences [28]. Pearson's correlation coefficient was also calculated to determine the strength and direction of the linear relationship between the results of the mF8WT and aF8WT, providing additional insights into their correspondence. The correlation coefficients were interpreted as negligible (<0.10), weak (0.10–0.39), moderate (0.40–0.69), strong (0.70–0.89), and very strong (0.90–1.00) correlations [30]. The results of the mF8WT and aF8WT were graded according to age and sex as suggested by the Korean National Fitness 100, respectively, and the agreement of the participants' grades between the two methods was evaluated by calculating the weighted Cohen's Kappa coefficient. The interpretation of kappa values followed standard thresholds: less than 0.20 as poor agreement, 0.21 to 0.40 as fair agreement, 0.41 to 0.60 as moderate agreement, 0.61 to 0.80 as substantial agreement, and greater than 0.80 as almost perfect agreement [31].

All statistical analyses were performed using SPSS (version 25.0; IBM Corp., Armonk, NY, USA) and MedCalc version, 19.4.1 (MedCalc Software, Ostend, Belgium), and statistical significance was set at $p < 0.05$.

## Results

### Baseline characteristics

The baseline characteristics of the participants are presented in Table 1. The total number of participants included in the analysis was 83, and the mean age of the participants was 71.6 ± 4.7 years. There were 14 males (16.9%) and 69 females (83.1%), and the mean BMI was 24.8 ± 2.9 kg/m². Based on the K-FRAIL scale, 60 (72.3%) participants were robust, 22 (26.5%) were pre-frail, and only 1 was frail (1.2%).

### Validity of the DSFT system for measuring the F8WT

The results of the mF8WT and aF8WT for each participant are presented in Table 2 and Fig 3. The mean of the participants' mF8WT results was 29.1 ± 4.9 seconds and that of the aF8WT

**Table 1. Baseline characteristics.**

| Variables | N = 83 |
|---|---|
| Age, years | 71.6 ± 4.7 |
| Sex (male/female, %) | 14 (16.9)/ 69 (83.1) |
| Height, cm | 155.4 ± 6.7 |
| Weight, kg | 59.9 ± 8.0 |
| BMI, kg/m² | 24.8 ± 2.9 |
| K-FRAIL, points | 0.33 ± 0.57 |
| 0 point | 60 (72.3%) |
| 1 point | 17 (20.5%) |
| 2 points | 5 (6.0%) |
| 3 points | 1 (1.2%) |

BMI, body mass index; K-FRAIL, the Korean version of the Fatigue, Resistance, Ambulation, Illnesses, and Loss of weight scale.

Based on the K-FRAIL score, frailty status is categorized as follows: robust (0), prefrail (1–2), and frail (≥3).

**Table 2. Comparison of the results of the manual or automated measurements of physical function.**

| Manual measurements | Automated measurements | Bland–Altman bias (95% LoA) | ICC | Correlation coefficient |
|---|---|---|---|---|
| 29.1 ± 4.9 s | 29.8 ± 4.9 s | −0.7 (−4.8 to 3.3) | 0.95 (0.91–0.97) | 0.91 |

LoA, limits of agreement; ICC, intraclass correlation coefficient.

results was 29.8 ± 4.9 seconds. The ICC value was 0.95 (0.91–0.97), which indicated excellent agreement, and this trend was also observed in the Pearson's correlation coefficient (r = 0.91, very strong correlation). In the Bland–Altman analysis, the 95% CI of the LoA was −0.7 (−4.8 to 3.3) seconds, and most of the data were distributed along the identity line in the scatter plot and heat map.

According to the grading of physical function by age and sex (very fast, fast, moderate, slow, and very slow) presented by the Korean National Fitness 100, 23 participants were classified as fast (27.7%), 51 as moderate (61.4%), and 2 as slow (10.8%) in the mF8WT and 18 as fast (21.7%), 49 as moderate (59.0%), 14 as slow (16.9%), and 2 as very slow (2.4%) in the aF8WT. Analysis of the agreement in grades between the mF8WT and aF8WT results yielded a Cohen's kappa coefficient of 0.56, with 71.1% (n = 59) of participants showing no difference in physical function grades between the two measurements. In addition, 22.9% (n = 19) showed that the aF8WT results were one grade slower and 6.0% (n = 5) showed that the aF8WT results were one grade faster than the mF8WT results. No differences were observed in more than two grades between the mF8WT and aF8WT results.

## Test-retest reliability of the DSFT system for measuring the F8WT

The results of the test-retest reliability of the automated measurements using the DSFT system are shown in Table 3 and Fig 4. Of the total participants, 27 (71.0 ± 4.5 years; 48 females [78.7%]) participated in the reliability test. The mean aF8WT results at the participants'

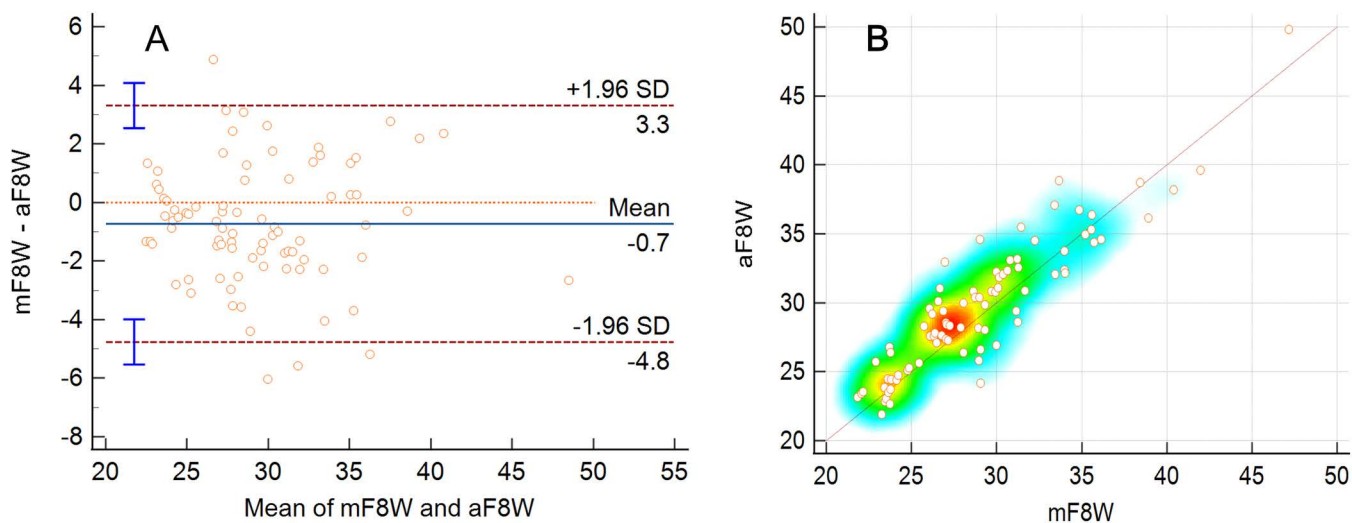

**Fig 3. Graphical presentations of the agreement between the manual and automated measurements of the figure-of-eight walk test.** (A) Bland-Altman plots. (B) Scatter plots and heat maps of all the data with the identity line.

**Table 3. Test-retest reliability of automated measurement using the DSFT system.**

| 1st Session | 2nd Session | Bland–Altman bias (95% LoA) | ICC |
|---|---|---|---|
| 30.9 ± 4.7 s | 29.6 ± 4.9 s | 1.2 (−2.6 to 5.1) | 0.94 (0.81–0.98) |

DSFT, digital senior fitness test; LoA, limits of agreement; ICC, intraclass correlation coefficient.

baseline (first session) was 30.9 ± 4.7 seconds, and the result at the second session, performed one week later, was 29.6 ± 4.9 seconds. In the Bland–Altman analysis, the 95% CI of the LoA was −1.2 (−2.6 to 5.1) seconds and the ICC was 0.94 (0.81–0.98, p < 0.001), showing excellent test-retest reliability.

## Discussion

In this study, the results of the aF8WT using the DSFT system showed excellent agreement with the results of the mF8WT traditionally performed by an assessor, and the error between the manual and automated measurements was within a clinically acceptable range, indicating a level of agreement comparable to or exceeding that observed between manual measurements conducted by different technicians. In addition, the aF8WT demonstrated a high level of test-retest reliability, confirming the feasibility of administering the aF8WT using the DSFT system to assess actual walking ability and to monitor changes in walking ability in community-dwelling older adults.

With the advancement of digital healthcare technology, several studies have been conducted to measure the physical function of older adults using digital devices. For example, the SPPB, which is commonly used to assess frailty in older adults, has already been evaluated in clinical settings using multi-sensor-based kiosks [22], and various wearable sensors are being used to analyze gait-related data to assess frailty and mobility impairment in older adults [32]. Additionally, the risk of falls and health-related outcomes in older adults can be predicted by analyzing data collected using various sensors on smartphones [33,34]. However, despite a thorough literature search, we were unable to find any reports of the F8WT being performed

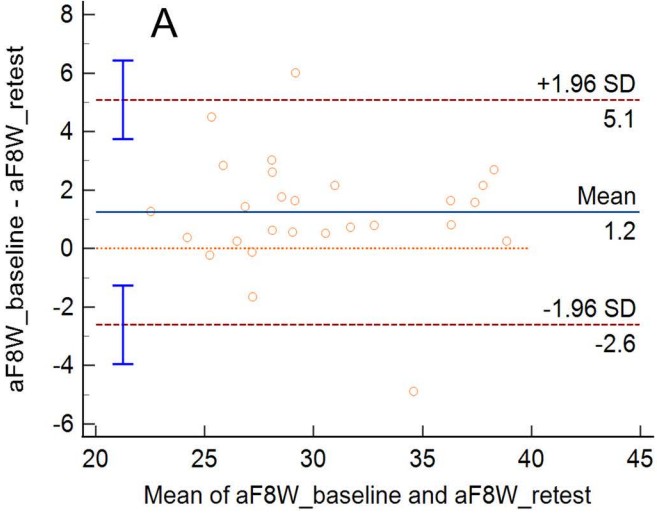
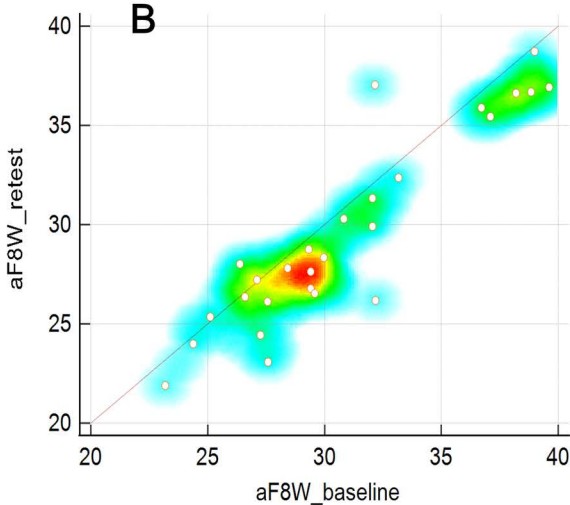

**Fig 4. Graphical presentations of the test-retest reliability of the automated figure-of-eight walk test.** (A) Bland-Altman plots. (B) Scatter plots and heat maps of all the data with the identity line.

using a digital device. In contrast to the SPPB or TUG test, which can already be measured using a digital device [22–24], the F8WT evaluates not only straight-path walking ability but also the ability to walk on a curved path, which is necessary for actual movement in daily life [15]. Therefore, the aF8WT by DSFT system can help assess the actual walking ability of community-dwelling older adults.

The results of the aF8WT using a digital device showed excellent agreement with those of mF8WT in this study. Because no previous study to our knowledge has evaluated the F8WT using a digital device, the results of the aF8WT using the DSFT system cannot be directly compared with the results measured using other digital devices. Additionally, comparing the results of mF8WT across studies is also challenging because there is currently no universally standardized protocol, leading to methodological differences between researchers. However, a previous study assessing the agreement between raters for the technician-administered manual measurement of the F8WT in community-dwelling older adults reported an ICC of 0.90 (0.71–0.97) [15], which is similar to our results (ICC = 0.95 [0.91–0.97]). In previous studies that measured walking ability using the TUG test with a video camera, which is similar to the DSFT system, the ICC of the automated measurement was > 0.90 [35]. The Korean National Fitness 100, which we used as a test protocol in our study, provides individualized exercise interventions according to the grade of the participant's physical function, and the automated measurement using the DSFT system classified the participants with the same grade as that measured manually in more than 70% of the cases. Based on these results, the aF8WT using the DSFT system is considered sufficiently applicable in clinical settings.

Monitoring physical function in community-dwelling older adults requires high test-retest reliability to ensure that changes in repeated physical function test results reflect actual changes in physical function rather than measurement errors, such as learning effects. In this study, the aF8WT using the DSFT system showed excellent test-retest reliability (ICC = 0.94, 0.81–0.98), confirming that it can be used to monitor changes in actual walking ability related to ADLs in older adults. Although a direct comparison is not possible because no previous studies to our knowledge have measured the F8WT using a digital device, a previous study of the test-retest reliability of manual measurement of the F8WT reported an ICC of 0.93 (0.85–0.97), with a mean difference of 1.4 seconds [17], which was similar to our study (ICC = 0.94 [0.81–0.98], with a mean difference of 1.2 seconds). Unlike manual measurement, which can only be performed by a physiotherapist, automated measurement using the DSFT system allows older adults to self-assess without the assistance of an assessor; therefore, it can be used in communities other than medical facilities to continuously monitor the actual walking ability of older adults.

This study had several limitations. First, it is difficult to generalize the results of this study to older adults who are cared for in nursing homes or those with severe physical frailty because the participants recruited for this study were community-dwelling healthy older adults. Second, previous studies have used slightly different methods to conduct the mF8WT [15–17], and this study was conducted using the method implemented in the Korean National Fitness 100 [26]; therefore, the validity and test-retest reliability of the aF8WT using other test protocol should be verified in the future. Finally, although the DSFT system can measure gait parameters such as step count and step length, we did not include these in our analysis, as their inclusion would have significantly expanded the scope of this study. We plan to validate these additional gait parameters in a further study to assess the clinical applicability of the DSFT system. Nevertheless, this study is valuable as it is the first to our knowledge to measure the F8WT using a digital device that can measure the walking ability required for actual ADLs.

## Conclusion

The results of the aF8WT measured using the DSFT system showed excellent validity and rest-retest reliability. The DSFT system is expected to measure and continuously monitor the walking ability of older adults living in the community to provide appropriate individualized interventions for physical frailty.

## Acknowledgments

The author has no acknowledgments to report.

## Author contributions

**Conceptualization:** Hyun-Ho Kong, Kwangsoo Shin, Dong-Seok Yang.

**Data curation:** Hyun-Ho Kong, Kwangsoo Shin, Dong-Seok Yang, Hyeon-Seong Joo.

**Formal analysis:** Dong-Seok Yang, Hyeon-Seong Joo.

**Funding acquisition:** Hyun-Ho Kong, Dong-Seok Yang.

**Investigation:** Hyun-Ho Kong, Hye-Young Gu.

**Methodology:** Hyeon-Seong Joo, Hyun-Chul Shon.

**Project administration:** Dong-Seok Yang.

**Resources:** Dong-Seok Yang, Hye-Young Gu.

**Software:** Kwangsoo Shin, Hye-Young Gu, Hyeon-Seong Joo.

**Supervision:** Hyun-Ho Kong, Hyun-Chul Shon.

**Validation:** Hyun-Ho Kong.

**Visualization:** Hyun-Ho Kong, Hye-Young Gu.

**Writing – original draft:** Hyun-Ho Kong.

**Writing – review & editing:** Hyun-Ho Kong, Hyun-Chul Shon.

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
