## [Decision Letter · Decision Letter 0]

20 Oct 2024

PONE-D-24-26140Digital assessment of walking ability: validity and reliability of the automated figure-of-eight walk test in older adultsPLOS ONE

Dear Dr. Kong,

Thank you for submitting your manuscript to PLOS ONE. After careful consideration, we feel that it has merit but does not fully meet PLOS ONE’s publication criteria as it currently stands. Therefore, we invite you to submit a revised version of the manuscript that addresses the points raised during the review process.

 In addition to the reviewer's comments, please consider the below: The manuscript is exploring the validity and reliability of the automated F8WT. The test itself is a functional, practical test that is performed with minimal equipment and therefore applicable in almost any environment. The proposed additions, however, make it costlier, more difficult to execute and without any apparent benefit. There is still a need for a supervisor (both for correct execution and safety of the participants), there is a need for additional equipment, with the main outcome still being time taken to complete the test and not additional information by the more sophisticated experimental set-up. Therefore, the conclusion of the test being of clinical significance and application is difficult to support.    It would be useful to offer a justification for the statistical analysis followed, both for the validity analysis (e.g. DOI: https://doi.org/10.1054/ptsp.1999.0001) and the reliability analysis (e.g. DOI: 10.2165/00007256-199826040-00002); for example, justification for the lack of bias assessment. Nonetheless, one aspect that must be discussed is the results in terms of the actual scores and the range they have produced. For example, how do the mean scores of ~30s compare to the literature (e.g. DOI: 10.1093/gerona/glaa035) and how sensitive / helpful a test can be with the wide 95%CI reported here? Finally, there appears to be an attempt at assessing agreement between the classification of participants by the two systems, but no statistical analysis or discussion is presented. The purpose of the manuscript make these discussions vital before the claim for clinical relevance and application can be made in the Discussion.  Ideally, specify the type of validity you are examining (or use the term 'method comparison'). Similarly, to avoid confusion, please use the relevant terms consistently (e.g. reliability vs agreement). 

We look forward to receiving your revised manuscript.

Kind regards,

Theodoros M. Bampouras

Academic Editor

PLOS ONE

Journal Requirements:

“This research was supported by the Medical Device Technology Development Program (grant number: 20014701, modular quantitative aging assessment and care service based on multiple sensors for aging in-home) funded by the Ministry of Trade, Industry, and Energy (MOTIE, Sejong, Republic of Korea).”

Reviewers' comments:

Reviewer's Responses to Questions

**Comments to the Author**

1. Is the manuscript technically sound, and do the data support the conclusions?

Reviewer #1: Yes

2. Has the statistical analysis been performed appropriately and rigorously? 

Reviewer #1: Yes

3. Have the authors made all data underlying the findings in their manuscript fully available?

Reviewer #1: No

4. Is the manuscript presented in an intelligible fashion and written in standard English?

Reviewer #1: Yes

5. Review Comments to the Author

Reviewer #1: The study is assessing the validity and reliability of a digital device to measure the figure-of-eight walk test in older adults. The study has a large sample size and has assessed both criterion validity and test-retest reliability.

A few questions that could help improve the paper:

1. A better justification for the use of digital assessment of the test in the introduction could help: For example, intra- and inter-rater reliability of manual testing of these tests, especially in multi-center studies, could be problematic, so a digital device could help.

2. Methods: The digital device procedure needs a better explanation. What are cameras specifically used for? It is said that AI is used. AI is specifically used for which task? Was ChatGPT or custom-made AI scripts used? Do you get the results right away?

3. Does the specific test count the number of steps and smoothness/accuracy of the test or just the time taken? This is often considered for this test.

4. Which ICC was used here? For absolute agreement or consistency? There are multiple ways (two-way random or fixed, and so forth) to perform ICC for reliability studies.

5. What is the cost of the digital device? Could that be a limitation? How long does it take to set it up and analyze compared to manual testing?

6. It is stated, "the error between the manual and automated measurements was within a clinically acceptable range" What is the clinically accepted range or what is the clinically meaningful threshold for the test?

6. PLOS authors have the option to publish the peer review history of their article (what does this mean? ). If published, this will include your full peer review and any attached files.

**Do you want your identity to be public for this peer review?** For information about this choice, including consent withdrawal, please see our Privacy Policy .

Reviewer #1: No

---

## [Author Response · Author response to Decision Letter 1]

14 Nov 2024

Dear Academic Editor and Reviewer,

We would like to extend our sincere appreciation for your thorough and insightful review of our manuscript, titled "Digital assessment of walking ability: validity and reliability of the automated figure-of-eight walk test in older adults." Your comments and questions have significantly contributed to enhancing the quality and clarity of our study.

In response, we have carefully addressed each of your suggestions and provided detailed explanations in the "Response to Reviewers" and "Response to Academic Editor" documents. Additionally, we have revised the manuscript to incorporate these changes, ensuring alignment with your recommendations. Key revisions include the following:

- Enhanced justification for the use of a digital device in measuring walking ability and expanded explanations on its benefits, especially concerning reliability in multi-center studies.

- Detailed descriptions of our statistical approach, with additional references for validity and reliability, and analysis adjustments as recommended.

- Clarification regarding the device setup, usage of AI technology, and specific operational details, which should provide more comprehensive context.

- Revisions to consistently apply terminology, particularly for validity, reliability, and agreement, to improve clarity for readers.

We hope that these changes address your concerns and improve the manuscript. Thank you again for your time and constructive feedback.

Kind Regards,

Hyun-Ho Kong & Hyun-Chul Shon

---

## [Decision Letter · Decision Letter 1]

4 Dec 2024

PONE-D-24-26140R1Digital assessment of walking ability: validity and reliability of the automated figure-of-eight walk test in older adultsPLOS ONE

Dear Dr. Kong,

Thank you for submitting your manuscript to PLOS ONE. After careful consideration, we feel that it has merit but does not fully meet PLOS ONE’s publication criteria as it currently stands. Therefore, we invite you to submit a revised version of the manuscript that addresses the points raised during the review process. It is clear that careful consideration of the comments took place and the manuscript has improved considerably. There are only some minor points that need addressing, please see below:

Introduction - what is the relevance of the 2nd paragraph, given that there is a test that measures curved path walk, the F8WTIntroduction - the premise re the rationale for the study needs some more development; for example, several studies (e.g. https://pubmed.ncbi.nlm.nih.gov/30657232/ , https://www.sciencedirect.com/science/article/abs/pii/S2211034822009348 , https://linkinghub.elsevier.com/retrieve/pii/S0031940618301330) all have reported good inter-rater agreement, albeit in clinical populations. This appears to dent the claim re minimising inter-rater reliability (although the reasons re time-efficiency etc still hold true).Please clarify that it is automated measure F8WT or manual measure F8WT throughout the manuscriptStatistical analysis - please clarify which tests are used for what for posterityCohen's kappa - was that weighted kappa (given there were more than two categories) and was there a threshold set for acceptable / good etc?

We look forward to receiving your revised manuscript.

Kind regards,

Theodoros M. Bampouras

Academic Editor

PLOS ONE

Journal Requirements:

Reviewers' comments:

Reviewer's Responses to Questions

**Comments to the Author**

1. If the authors have adequately addressed your comments raised in a previous round of review and you feel that this manuscript is now acceptable for publication, you may indicate that here to bypass the “Comments to the Author” section, enter your conflict of interest statement in the “Confidential to Editor” section, and submit your "Accept" recommendation.

Reviewer #1: All comments have been addressed

2. Is the manuscript technically sound, and do the data support the conclusions?

Reviewer #1: Yes

3. Has the statistical analysis been performed appropriately and rigorously? 

Reviewer #1: Yes

4. Have the authors made all data underlying the findings in their manuscript fully available?

Reviewer #1: Yes

5. Is the manuscript presented in an intelligible fashion and written in standard English?

Reviewer #1: Yes

6. Review Comments to the Author

Reviewer #1: (No Response)

7. PLOS authors have the option to publish the peer review history of their article (what does this mean? ). If published, this will include your full peer review and any attached files.

**Do you want your identity to be public for this peer review?** For information about this choice, including consent withdrawal, please see our Privacy Policy .

Reviewer #1: No

---

## [Author Response · Author response to Decision Letter 2]

6 Dec 2024

We sincerely thank the reviewers for their insightful and constructive feedback on our manuscript. Based on these comments, we have revised the manuscript to enhance its clarity, quality, and overall scientific value. Below is a summary of the revisions made:

- Clarified the relevance and purpose of the second paragraph in the Introduction, focusing on the unique contributions of the Figure-of-Eight Walk Test (F8WT) and its automated implementation.

- Revised the manuscript to address the rationale for minimizing inter-rater reliability and emphasized the advantages of using automated F8WT (aF8WT) in diverse settings.

- Clearly specified whether the F8WT referred to automated (aF8WT) or manual (mF8WT) in all relevant sections of the manuscript.

- Expanded the Statistical Analysis section to explicitly state the purpose of each statistical test and its connection to the study objectives.

- Clarified the use of weighted Cohen’s kappa for ordinal data and included interpretation thresholds for kappa values.

For detailed responses to specific comments, please refer to the attached "Response to Reviewers" document.

---

## [Editor Report · Decision Letter 2]

15 Dec 2024

Digital assessment of walking ability: validity and reliability of the automated figure-of-eight walk test in older adults

PONE-D-24-26140R2

Dear Dr. Kong,

We’re pleased to inform you that your manuscript has been judged scientifically suitable for publication and will be formally accepted for publication once it meets all outstanding technical requirements.

Kind regards,

Theodoros M. Bampouras

Academic Editor

PLOS ONE
---

## [Editor Report · Acceptance letter]

PONE-D-24-26140R2

PLOS ONE

Dear Dr. Kong,

I'm pleased to inform you that your manuscript has been deemed suitable for publication in PLOS ONE. Congratulations! Your manuscript is now being handed over to our production team.

Kind regards,

on behalf of

Dr. Theodoros M. Bampouras

Academic Editor

PLOS ONE